# From Adipose Tissue to Endothelial Cells—Pleiotropic Role of Vaspin in Pathogenesis of Metabolic and Cardiovascular Diseases

**DOI:** 10.3390/biomedicines13123040

**Published:** 2025-12-10

**Authors:** Krzysztof Maksymilian Dąbrowski, Hubert Mateusz Biegański, Anna Różańska-Walędziak

**Affiliations:** 1Medical Faculty, Collegium Medicum, Cardinal Stefan Wyszynski University in Warsaw, 01-938 Warsaw, Poland; krzmaksdab@gmail.com (K.M.D.); ski.bieg.an@gmail.com (H.M.B.); 2Department of Human Physiology and Pathophysiology, Faculty of Medicine, Collegium Medicum, Cardinal Stefan Wyszynski University in Warsaw, 01-938 Warsaw, Poland

**Keywords:** vaspin, serpinA12, metabolic syndrome, diabetes mellitus, atherosclerosis, hypertension

## Abstract

**Background:** Vaspin (also known as serpinA12) is a recent discovery among adipokines. It plays a significant role in obesity-related conditions, many of which are classified as chronic, inflammatory or lifestyle diseases. Due to its anti-inflammatory and insulin-sensitizing properties, vaspin has been investigated as a biomarker and potential therapeutic agent. **Methods:** A literature review was conducted using the MEDLINE and SCOPUS databases using the phrases “vaspin” and “serpinA12” to summarize the most recent and influential research concerning vaspin’s mechanisms and influence on various tissues. **Results and Conclusions:** Vaspin is notably involved in metabolic syndrome, and it is generally associated with mitigating conditions like insulin resistance and obesity-related chronic inflammation. In addition, its beneficial effects on endothelial and smooth muscle cells under hyperglycemic and hyperlipidemic conditions are also well documented. There is growing evidence that vaspin positively impacts cardiovascular health, reducing the risk of ischemic stroke and the development of atherosclerosis. Moreover, some studies suggest a direct influence of vaspin on the central nervous system, with its administration shown to reduce the expression of neuropeptide Y, a key regulator of food intake. Many of the reviewed sources highlight vaspin not only as a possible biomarker but also as a promising therapeutic candidate. However, despite intensive research on vaspin over the past 20 years, there are significant disparities between animal and in vitro studies versus human studies. A further limitation in the field is the lack of standardization in research methodologies, which contributes to inconsistent and sometimes contradictory results.

## 1. Introduction

Vaspin, also called visceral adipose tissue-derived serpin or simply serpinA12, is a newly discovered adipocytokine—a member of the serine protease inhibitor (serpin) family [1,2]. It was isolated for the first time in 2005 from the visceral white adipose tissue (WAT) of Otsuka Long-Evans Tokushima Fatty (OLETF) rats [1]. This particular rat strain is known for the spontaneous development of type 2 diabetes mellitus (T2DM), hypertension, hyperglycemia and central obesity [1,3]. An increase in serum levels and in the adipose tissue expression of vaspin was observed, peaking around the 30th week of the rats’ lives, when hyperglycemia and insulin resistance were most pronounced [1,2]. Additionally, both pioglitazone and insulin administration normalized vaspin expression and serum concentration [1,2]. The administration of recombinant human vaspin to mice fed a high-fat, high-sucrose diet significantly lowered serum glucose levels without affecting serum insulin concentrations, leading to the conclusion that vaspin has insulin-sensitizing and anti-inflammatory activity [1,2].

Vaspin is expressed in a number of human tissues, including subcutaneous (SAT) and visceral adipose tissue (VAT) [4,5], the liver and the pancreas [6], the skin [7] and the placenta [8]. The newly discovered adipokine’s affiliation with the broad serpin family and its multidirectional anti-inflammatory effects attracted widespread interest from scientists, who recognized vaspin as a potential new target for research into the treatment of metabolic and obesity-related diseases [2]. Vaspin has a comprehensive effect on various tissues of the human body, which has been graphically presented in Figure 1 and is discussed further in this review.

There are, however, several obstacles in current vaspin research, including the lack of standardization and the considerable number of contradictory studies, which may postpone any practical application of this adipokine. These limitations underscore the need for more unified research approaches. Therefore, this review aims to summarize the current state of knowledge on vaspin and to identify the key challenges that must be addressed to advance future research.

## 2. Molecular Basis of Vaspin’s Action, Its Cofactors and Intracellular Pathway Mediation

### 2.1. The Serpin Activity of Vaspin

Consisting of 395 amino acids, vaspin shows a significant homology with α_1_-tripsine and has a typical structure for serpins—three β-sheets (A, B and C), nine α-helices and a flexible reactive center loop (RCL) with a protease recognition center [1,9]. It is worth noting that the central β-sheet A is composed of five β-strands [9]. Interestingly, vaspin exhibits remarkable thermostability compared to other members of the serpin family, maintaining its activity at 60 °C [10].

The vaspin-mediated inhibition process begins when the protease recognizes a specific sequence within the RCL, initiating the formation of a non-covalent interaction between the RCL and the protease’s active site, along with additional stabilizing interactions at regions outside the active site, known as exosites [11]. During the initial acylation reaction, the RCL moves and incorporates itself into the central β-sheet A, becoming the sixth strand [12]. Simultaneously, the relative position of protease and serpin shifts. As a consequence of the inhibition mechanism, the protease’s reactive center becomes severely distorted, permanently preventing dissociation of the serpin–protease complex [13,14]. Moreover, possible distortions in more distal parts of the protease render the whole complex prone to enzymatic degradation in proteasomes [15]. Unfortunately, even though there is strong evidence of vaspin’s beneficial effect on glucose homeostasis [16], the exact mechanisms and action pathways are not fully understood [17].

Despite the fact that vaspin does not inhibit commonly occurring proteases such as collagenase, urokinase, elastase, trypsin and factor Xa [1], it does have an anti-inflammatory effect on tissues by inhibiting kallikrein 7 (KLK7) and kallikrein 14 (KLK14) [18,19]—two members of the broader serine protease family, exhibiting chymotrypsin-like and combined chymotrypsin- and trypsin-like activities, respectively [20,21,22]. Both KLK7 and KLK14 are known for taking part in the desquamation of the stratum corneum by digestion of corneodesmosomes and being involved in the pathogenesis of skin diseases such as Netherton syndrome, atopic dermatitis, psoriasis and melanoma [23,24,25,26,27] Regarding KLK7, its expression in murine pancreatic β-cells is also well established [18]. On a molecular level KLK7 causes cleaving and degradation of both insulin A and B chains [18,21,28] and propagates infiltration and activation of inflammatory macrophages, particularly in epididymal adipose tissue [29]. Thus, KLK7 activity is thought to be a factor in the pathogenesis of systemic inflammation and insulin resistance in the course of obesity [29]. Heiker et al. observed that vaspin directly diminishes insulin degradation in blood plasma and pancreatic islands, which suggests that the anti-diabetic effect of vaspin results from insulin level stabilization rather than its direct insulin-sensitizing influence on target tissues [18]. However, in light of recent research, it is not definite, as Liu et al. reported a direct insulin-sensitizing effect of vaspin on pancreatic islet β cells [30]. These discrepancies between scientific findings highlight the need for further research to clearly define the relationship between the vaspin–KLK7 axis and the progression of insulin resistance.

### 2.2. The Effects of Auxillary Molecules’ Binding on Vaspin’s Activity

Similarly to other serpins, vaspin binds glycosaminoglycans (GAGs) that serve as cofactors in the inhibition reaction, bridging the serpin and protease particles. Such binding induces the formation of a Michaelis complex and consequently precipitates the inhibition process [11]. Heparin is a characteristic cofactor of vaspin, to which it binds with high affinity, exhibiting a dissociation constant of 21 nM. At the same time, vaspin does not bind any other GAGs, which makes it unique among known serpins [31]. The heparin-binding capacity of vaspin is strongly influenced by genetic factors; mutations at the basic residues K359 and R211 decrease heparin affinity and heparin-mediated enhancement of the inhibition reaction by up to 40% [31]. The presence of heparin moderately enhances the inhibition of KLK7; however, this is not the case for KLK14, where heparin does not accelerate the inhibition process and instead interferes with the formation of the serpin–protease complex [19,31]. In order to determine whether this heparin-binding mechanism has relevance in physiological conditions, a study on HaCaT cells was conducted [32], revealing that a significant concentration of vaspin can be found in the extracellular matrix (ECM). Elevated levels of vaspin were also detected in the membrane fractions of liver tissue lysates from transgenic vaspin mice, where it co-localized with α2-macroglobulin, a key inhibitor of ECM-degrading matrix metalloproteases [33].

Another pivotal cofactor of vaspin is the 78 kDa glucose-regulated protein (GRP78), an endoplasmic reticulum (ER) chaperone involved in the unfolded protein response [34], whose association with vaspin was identified in HepG2 liver cells [33]. In contrast to heparin, GRP78 does not have any role in the protease interaction. Nakatsuka et al. discovered that GRP78 takes part in tissue-specific signaling pathways mediation together with Dnaj-like MTJ1 protein in liver tissue [33] and with the voltage-dependent anion channel (VDAC) in human aortic endothelial cells [35]. The interaction between MTJ1 and GRP78 is essential for the transport of GRP78 from the endoplasmic reticulum to the cell surface [36], where it mediates intracellular signaling using phosphoinositide 3-kinase (PI3-K), protein kinase B (Akt), 5′ AMP-activated protein kinase (AMPK) and nuclear factor kappa-light-chain-enhancer of activated B cells (NF-κB) [37]. In this way vaspin exerts influence on various tissues: it ameliorates ER stress in obesity, improves both glucose and lipid metabolism in hepatocytes via Akt and AMPK [33] and attenuates steatosis-induced fibrosis in hepatic stellate cell lines via AMPK [38]. Additionally, in endothelial cells, GRP78 together with VDAC functions as a receptor for kringel 5 (K5)—a plasminogen-derived proteolytic fragment that inhibits angiogenesis and induces apoptosis in proliferating endothelial cells by increasing intracellular Ca^2+^ levels [39,40]. Vaspin bound to GRP78 effectively counteracts the pro-apoptotic effect of K5 by inducing Akt phosphorylation and subsequently inhibiting calcium ion influx [35]. Nonetheless, at the same time, elevated K5 levels disrupt the binding of vaspin to GRP78 while influencing intracellular Ca^2+^ through VDAC [41].

The recent 2025 discovery made by Möhlis et al.—namely, that internalized vaspin has a DNA-binding property in the nucleus of the stroma-vascular fraction of subcutaneous adipocytes and human coronary artery endothelial cells—is particularly interesting. Via the LRP1 receptor, vaspin can be internalized and bind both single- and double-stranded DNA molecules, further enhancing the inhibition of KLK7 [42,43]. This makes vaspin only the second member of the serpin family known to possess DNA-binding properties [44]. Vaspin does not appear to exhibit sequence-specific DNA–protein interactions and binds DNA at a site distinct from those for heparin and polyphosphates [45], as vaspin mutants lacking these residues demonstrated comparable enhancement of KLK7 inhibition [43].

All of the mechanisms described above indicate a significant tendency of vaspin to interact with plasma membranes. Moreover at least some of pathway activations caused by vaspin may be mediated through binding to ECM GAGs or via GRP78 and other cell-surface molecules. Thus, studies employing non-heparin vaspin mutants, together with a deeper understanding of the vaspin-GRP78 interaction, will be essential for elucidating these mechanisms [17]. Although significant progress has been made in explaining the mechanisms by which vaspin affects various tissues, much remains to be discovered—including potentially unexpected functions such as DNA-binding enhancement.

The fundamental molecular mechanisms of vaspin, along with selected pathways discussed in the following sections, are summarized in Figure 2.

## 3. Vaspin in Obesity-Related Diseases

### 3.1. Obesity and Metabolic Syndrome

Over the past few decades, overweight and obesity have reached the status of global epidemic and major civilization-level health crisis [46]. These conditions are defined as excessive body mass accumulation, typically assessed using body mass index (BMI), and are diagnosed in patients exceeding 25 kg/m^2^ for overweight and 30 kg/m^2^ for obesity [47]. According to World Health Organization data, in 2022 approximately 43% of adults worldwide were struggling with excess weight and 26% were obese [47]. This problem is considerably more pronounced in developed countries; for instance, in the United States, 42% of adults are classified as obese and 73% as overweight [48,49].

A surplus of body mass is known to be a cause for many different health issues, such as higher risk of developing T2DM, increased chance of cardiovascular incidents, chronic lung diseases and even cancer [47,50]. For this reason, adipokines represent an attractive research direction, particularly those exhibiting protective properties, such as vaspin [1].

Metabolic syndrome (MetS), closely associated with obesity due to their co-occurrence, has been rapidly emerging in the past few decades and has currently reached the status of epidemic [51,52]. MetS is characterized by the simultaneous development of obesity, insulin resistance, dyslipidemia and hypertension, where every factor increases the risk of another’s progression [53]. The diagnosis of metabolic syndrome is established when at least three out of five of the criteria presented in Table 1 are met.

One of the very first observations in this context was that the administration of vaspin normalized the expression of genes associated with MetS, including GLUT4, resistin, adiponectin and leptin genes [1]. Nevertheless, vaspin’s relationship with MetS appears to be mediated primarily through insulin resistance, which is further discussed in the following sections [54].

As previously mentioned, vaspin was found to be associated with extensive body mass in OLETF rats, an established animal model for human metabolic syndrome [3,55]. Similarly, both serum vaspin concentrations and mRNA expression are positively correlated with body mass in humans. Vaspin is not only upregulated in patients who are overweight or obese [2,4,56,57,58] but also down-regulated in underweight individuals [59,60]. Additionally, the presence of the vaspin single nucleotide polymorphism rs2236242 allele A is associated with a 34% reduced risk of both obesity and MetS [61,62]. However, there are many contradictory studies regarding body mass changes. Some of them showed that vaspin serum levels increase following weight loss induced by exercise [63,64], while others observed opposite results [65,66,67]. A more confident finding is that physical activity alone, without changes in body mass, does not influence vaspin tissue or plasma levels, despite reducing cytokines such as IL-6 or TNF-α [68,69,70]. These findings highlight the dominant role of adipose tissue mass in vaspin expression and production, which aligns with the discovery that vaspin is expressed only in 23% of VAT, with a higher percentage in obese patients [4]. Moreover, overexpression of vaspin in transgenic mice resulted in a reduction of diet-induced weight gain, hyperinsulinemia and high glucose and cholesterol levels [71], even though administration of this adipokine had no effect on current body weight in already obese rats [30]. Despite these associations, vaspin likely influences body mass through modulation of insulin resistance in a manner similar to MetS [72,73].

### 3.2. Dependencies Between Vaspin Levels and Lipid Imbalance

It is well established that different types of food have a diverse influence on the serum lipid profile. It has also been noted that both lipid profile and a diverse diet are correlated with vaspin levels. Meta-analysis including 7446 patients showed a significant negative correlation between serum vaspin levels and factors such as plasma triglycerides, total cholesterol and LDL-cholesterol levels [74]. In addition, vaspin treatment in a mouse model of diabetes resulted in a reduction of circulating free fatty acid levels, while cholesterol remained unchanged [74]. Meanwhile, transgenic mice overexpressing vaspin were protected from increases in triglycerides and had slightly reduced LDL-cholesterol levels, despite being fed a high-fat, high-sugar diet [33]. In humans, serum vaspin showed a positive correlation with triglycerides and a modest correlation with LDL-cholesterol [63,75,76]. This highlights another compensative aspect of vaspin, as it not only ameliorates lipid profiles but also plays a protective role against high-lipid-induced apoptosis [77], an important factor in further-discussed atherogenesis. Regarding HDL-cholesterol, the available studies are too contradictory to draw a definitive conclusion about its correlation with vaspin [63,74,76]. One of the possible mechanisms improving lipid metabolism and providing protection from dyslipidemia is the reduction of uncontrolled triglyceride catabolism through amelioration of B-adrenergic activation of lipolysis [78]. Brown adipose tissue (BAT) also play a role in vaspin-level regulation in rats, as the expression of this adipokine is induced by an obesogenic diet and cold exposure, not by adrenergic activation, which may suggest cold exposure and expanding BAT activity could be potential weapons against MetS [79,80]. Additionally, patients with fenofibrate-treated hypertriglyceridemia have greatly increased vaspin serum levels and adipose tissue expression, excluding SAT [81].

### 3.3. Effects of Bariatric Surgery on Vaspin Plasma Levels

Bariatric surgery is a procedure during which the upper gastrointestinal tract is surgically altered to enhance satiety and diminish nutrient absorption. It is one of the few effective methods of obesity treatment and an important adjunct in managing MetS or T2DM [82,83]. What is important is that we change the hormonal balance of the bowels; therefore, it could also affect vaspin function [84]. In the short term (6 weeks) after restrictive surgery, vaspin serum levels decreased alongside mass reduction [85]. This observation confirms the correlation between vaspin and body mass, as HOMA-IR values remained unchanged. It may also serve as further evidence of vaspin’s compensatory function—following weight reduction, a lower adipokine level may be sufficient to maintain a given degree of insulin sensitivity. Researchers also suggested that decreased dietary intake may be responsible for the vaspin reduction [85]. These observations align with a longer-lasting study on patients who underwent Roux-en-Y gastric bypass surgery. After a 12-month observation period, serum vaspin levels decreased and were significantly correlated with a reduction in HOMA-IR values [86]. This observation contradicts that of a previously mentioned study. However, this is most likely due to the longer observation period, as the organism requires time to restore metabolic balance, including insulin sensitivity. Similar changes in vaspin levels, positively correlating with HOMA-IR values, were observed after laparoscopic vertical banded gastroplasty [87]. Additionally, high vaspin concentration was highlighted as a potential predictive factor in better post-bariatric surgery outcomes, as patients with higher serum vaspin concentrations improved their HOMA-IR values, insulin levels and fasting glucose concentration slightly faster [88]. Based on the current state of research, it is difficult to determine whether a significant direct relationship exists between gastrointestinal hormones and vaspin in the context of bariatric surgery.

### 3.4. Vaspin’s Influence on Adipocyte Metabolism and Growth

Vaspin not only helps prevent apoptosis [77] but also affects the cell proliferation and lipid accumulation processes. Administration of vaspin in hyperglycemic rats inhibited vascular smooth muscle growth by 40% [35,89,90] and ameliorated healing in the injured arteries of obese rats [35]. The cardiovascular associations of vaspin are discussed in detail in a separate section of this article. The most apparent changes in obesity are changes in body mass and fat tissue volume. Vaspin appears to inhibit adipocyte growth in WAT and reduce lipid accumulation in hepatocytes [33,71]. The direct effect of vaspin on lipid accumulation in adipocytes remains controversial. Some studies on cell cultures suggest that lipid accumulation is stimulated in a dose-dependent manner [91], whereas others do not confirm this phenomenon [78,92]. Dose-dependency is noteworthy, as in low vaspin concentrations of (1 nM; 45.2 ng/mL) preadipocytes enhance their metabolic and mitochondrial activity, while in high concentrations (100 nM; 4520 ng/mL) their differentiation is slowed [78,91]. This highlights the complex nature of vaspin, demonstrating that its actions extend beyond a simple on/off mechanism.

### 3.5. Type 2 Diabetes Mellitus

As mentioned earlier, when vaspin was first discovered and characterized back by Hida et al. in 2005, its action was primarily associated with a reduction in insulin resistance [1]. This beneficial effect prompted the question of whether vaspin might play a role in the development of type 2 diabetes mellitus and how it could potentially be used for therapeutic purposes.

Diabetes mellitus is a disease in which the body cannot regulate blood glucose levels, which in turn causes constant hyperglycemia [93]. As a result, many tissues and organs are damaged, often irreversibly. The vessels, kidneys, eyes and many more are affected in the course of the disease [93]. T2DM develops because of general inflammation and insulin resistance. As a result, pancreatic β-cells become overburdened and gradually deteriorate, leading to insufficient insulin secretion [93]. T2DM is a global health problem; in 2022, approximately 14% of adults were living with the disease, a figure that has doubled since 1990 [94]. We can diagnose diabetes mellitus when fasting glucose is ≥126 mg/dL, random plasma glucose is ≥200 mg/dL or HbA1c is ≥6.5% [93].

Numerous studies have reported a positive correlation between T2DM and vaspin [95,96], while an equal number have reported contradictory findings [97,98]. This issue remained controversial until 2014, when a meta-analysis of 1570 patients conducted by Feng et al. reported significantly higher serum vaspin levels in individuals with T2DM [56], in contrast to findings in rats, where vaspin levels were decreased [99]. A high vaspin level is a risk factor for T2DM [100]. In accordance with the previously established relationship between vaspin and insulin resistance, it has been demonstrated that vaspin levels are also elevated in gestational diabetes [101]. Although the vaspin single nucleotide polymorphism rs2236242 is associated with obesity and MetS, a meta-analysis found no correlation with T2DM [62]. As mentioned before, in the development of T2DM, inflammation and insulin resistance are the main culprits. Vaspin mitigates these factors through multiple mechanisms; however, its levels decline over the course of T2DM [102,103].

Low-grade systemic inflammation, a key driver of T2DM progression, can be assessed via inflammatory cytokines such as IL-1β, IL-6 and TNF-α [104], and vaspin seems to reduce the expression of all of them [32]. IL-6 was speculated to be a key mediator in T2DM and was proven to contribute to disease development by increasing insulin resistance through the induction of SOCS-3 gene expression [105]. However, large-scale meta-analyses suggest that this role is likely modest [106]. Nevertheless, IL-6 alongside vaspin was positively correlated with micro and macrovascular complications in diabetes [107]. TNF-α is another important inflammatory cytokine, which intensifies insulin resistance in T2DM by impairing insulin signaling through serine phosphorylation [108,109]. Administration of vaspin was proven to reduce TNF-α expression [1]. In rat vascular smooth muscle cells, injuries were attenuated through the inhibition of NF-κB and protein kinase C theta (PKCθ) [110]; however, treatment with vaspin did not alter the response of human umbilical vein endothelial cells to TNF-α–induced damage or the activation of JNK and p38 [111]. Nevertheless, NF-κB plays a significant role in the context of vaspin function and T2DM [112], as its activation seems to be a key factor in β-cells and whole islets inflammation [112]. Moreover, its expression is stimulated by hyperlipidemia and hyperglycemia, creating a “doom loop” in T2DM and MetS [113]. Through this molecular pathway, vaspin reduces, among others, the expression of TNF-α and IL-6, consequently reducing systemic inflammation [92]. Treatment with vaspin reduces inflammation in pancreatic β-cells, thereby improving their function and enhancing insulin secretion, in part through NF-κB inhibition [30]. Another way vaspin acts is directly through GPR78 receptors. It exerts an anti-inflammatory effect through GRP78/MTJ-1 in the mouse liver [33], GRP78 anion channels in endothelial cells [35] and GRP78 in many tissues of the ovaries [114]—these are all important elements in T2DM health consequences.

As mentioned before, vaspin also has a direct effect on the pancreas. It has been established that it is expressed in pancreatic islets of both rodents and humans [18,30]. Rat insulinoma cell line (INS-1) treated with vaspin upregulated IRS-2, which is a receptor responsible for mediating the insulin signal via the PI3K/Akt mTOR/p70S6K pathway. Through this molecular pathway cell proliferation is promoted, which can relieve insulin resistance in β-cells [30]. To assess the importance of IRS-2, knockout mice exhibiting a pronounced diabetic phenotype were created [115]. After vaspin administration INS-1 cells increased insulin secretion [30]. On top of that, vaspin prolonged the insulin half-time by inhibiting KLK7, which naturally helps mitigate insulin resistance [18]. It is also noteworthy that both continuous subcutaneous insulin infusion and acute bolus administration reduce vaspin concentrations while improving β-cell function in T2DM [116,117,118]. This phenomenon may account for the inconsistency between studies, as patients receive insulin therapy in varying regimens. Molecular connections between vaspin and T2DM were described in depth by Liu et al. [30].

### 3.6. Polycystic Ovary Syndrome

Polycystic ovary syndrome (PCOS) is a common endocrine disorder with a prevalence of up to 26% among women. It is characterized by hyperinsulinemia, steroid imbalance and irregular menstruation [119]. Given its strong association with MetS and insulin resistance, vaspin seemed like an interesting research direction. Serum vaspin concentrations turned out to be significantly higher in women struggling with PCOS [120,121,122]. Interestingly, there is no significant difference between serum vaspin concentrations of obese PCOS patients compared to non-obese PCOS ones, supporting the notion that vaspin primarily acts via insulin resistance, which may be similarly dysregulated in both PCOS and obesity [73,120], despite the fact that vaspin expression is elevated in the ovaries of obese individuals [123]. Additionally, increased vaspin seems to be a marker of increased diabetogenic and atherogenic risk in PCOS patients [121], as well as problems with ovulation [124]. Unfortunately, there are many contradictory studies considering metformin treatment. Vaspin levels increase after metformin treatment in rats [8], whereas in humans the effect appears inconsistent—some studies report reduced vaspin concentrations [98,122,125], others describe an increase [126] and some find no significant effect at all [120]. Leptin administration does not affect vaspin concentration in humans [127], despite its reversible effect on fasting-induced vaspin decline in rats [8].

## 4. Vaspin in a Wider Context

### 4.1. Vaspin Influence on Cardiovascular System Disorders

Cardiovascular disease continues to be the leading cause of death globally; according to World Health Organization 2022 data, an estimated 19.8 million people died from cardiovascular diseases, with 85% of deaths resulting from cardiac infarction and ischemic stroke [128]. These statistics worsen with each passing decade due to changes in modern society, including the increasing proportion of sedentary occupations, consumerism- and technology-focused culture and reduced time for recreation, all of which contribute to widespread physical inactivity and a high intake of sugars and saturated fats from highly processed foods [129]. All of this inevitably leads to increased incidence of atherosclerosis and other metabolic-associated diseases, such as metabolic syndrome, T2DM and hypertension, involved in the pathogenesis of cardiovascular diseases [130,131,132]. Vaspin appears to have a broad influence on the proper functioning of the circulatory system, most notably by protecting endothelial and smooth muscle cells and preventing ischemic stroke and the development of atherosclerosis. 

Atherosclerosis is a pathological process affecting the arteries, characterized by the formation of atherosclerotic plaques that may lead to reduced or obstructed blood flow in the affected vessels [133]. One of the key factors in the pathogenesis of atherosclerosis is endothelial dysfunction—a collection of processes involving pro-inflammatory cytokines that drive the transition from early atherogenesis to advanced vascular occlusive disease and infarction [134]. Although initial studies did not support an inhibitory effect of vaspin on TNF-α–induced intracellular pathways [111], subsequent research has demonstrated that vaspin plays a key role in preventing endothelial damage in obesity and metabolic disorders by mitigating the effects of chronic inflammation. Vaspin was shown to prevent a TNF-α-mediated inflammatory response in both cultured rat vascular smooth muscle cells and human aortic endothelial cells by inhibiting TNF-α-induced activation of NF-κB and consequently activation of PKCθ and the cell adhesion molecules ICAM-1, VCAM, E-selectin and MCP-1, thereby reducing platelet and monocyte adhesion [110,135]. Moreover, it has been established that vaspin’s inhibition of NF-κB is mediated solely by AMPK activation, being independent of Akt and eNOS—two other pathways influenced by vaspin in the vascular endothelium [77,135,136]. The systemic inflammation might be further mitigated by reduced expression of IL-1 and IL-6 in human endothelial cell line EA.hy926 [137].

As a key mediator of endothelium-dependent vasorelaxation, reduced nitric oxide synthesis by eNOS is a key component of endothelial dysfunction during the progression of atherosclerosis [138]. By stimulating STAT3 and consequently activating the expression of the dimethylarginine dimethylaminohydrolase (DDAH) II gene, vaspin suppresses asymmetric dimethylarginine (ADMA) levels, thereby boosting eNOS activity and increasing the bioavailability of nitric oxide (NO) [136]. Additionally, vaspin induces eNOS expression using the PI3-Akt pathway in endothelial progenitor cells under hyperglycemic conditions, effectively counteracting the suppressive effect of hyperglycemia on NO synthesis. By inhibiting acetylcholine esterase, vaspin augments the acetylcholine-dependent vessel relaxation mediated by NO [139].

Essential hypertension is diagnosed when systolic blood pressure exceeds 130 mmHg or diastolic blood pressure exceeds 80 mmHg. Most hypertension cases are idiopathic, but it has been suggested that salt intake and genetic response play a big part in its development [140]. As a component of MetS, vaspin may also contribute to its pathogenesis. Investigating the causes of hypertension is important, as it is a major contributor to the progression of cardiovascular diseases and affects up to 60% of individuals over 60 years of age [140]. Vaspin’s correlation with hypertension in humans is controversial, as there are only a few studies on the matter and most of them are contradictory [141,142]. Vaspin has been shown to exert protective effects against elevated blood pressure, as pre-treatment prevented the development of spontaneous hypertension [143] and pulmonary HT in rats [144]. Vaspin increases NO bioavailability in isolated blood vessels by roughly 60%, which could suggest it plays a role in ameliorating hypertension [136]. However, the reactivity of these arteries remains unchanged [143]. Another key factor of vaspin’s protective properties could be its inhibition of arterial mesenteric wall hypertrophy, most likely via reduction of TNF-α and reactive oxygen species synthesis [143], which is an additional factor in sustaining hypertension. Use of losartan and ramipril has no impact on vaspin serum concentration, which suggests that vaspin action is not mediated via the renin–angiotensin–aldosterone system [145].

All of the aforementioned research comes to the same conclusion: vaspin has a significant protective effect on blood vessels. Unfortunately, there is a large disparity between the number of in vitro and in vivo studies on animals. However, the existing ones allow us to reach the same conclusion. Lin et al. reported that the progression of atherosclerotic plaques in apolipoprotein E–deficient mice, which are prone to developing atherosclerosis, was greatly inhibited in a research sample infected with vaspin-encoding lenti-virus [146]. The resulting vaspin expression suppressed endoplasmic reticulum stress–induced macrophage apoptosis, a key process contributing to advanced plaque necrosis [146,147].

Although reported vaspin levels in patients with atherosclerosis vary among studies, they are consistently higher than in healthy individuals [148,149]. Considering the beneficial effects of vaspin described above, it is plausible that this adipokine functions as a compensatory factor in vascular diseases. Vaspin has been shown to correlate with various cardiovascular diseases, although the nature of these associations appears to differ depending on the specific condition. Elevated vaspin concentrations have been observed in patients with coronary artery disease [148,150], whereas lower levels are reported in individuals with atherosclerosis of the coronary arteries [151]. In cases of severe ischemic stroke, vaspin levels demonstrate an inverse correlation with disease severity [152,153,154]. Similarly, reduced vaspin concentrations have been documented in patients experiencing myocardial infarction [155,156]. These findings suggest that vaspin may serve as a potential predictor of acute cardiovascular events, reflecting its possible role in both disease progression and the body’s compensatory response.

### 4.2. Vaspin Effects on Satiety

Another perspective on the role of vaspin in obesity is its impact on appetite perception and eating behavior. Administration of vaspin into the peritoneum or via intravenous injection resulted in a significant reduction in food intake in rodents, which was temporary and lasted for about one day [157]. On the other hand, intracerebral administration of vaspin caused the same effect, lasting much longer—even up to five days [157,158,159]. This phenomenon seems to be mediated by decreased gene expression of the orexigenic neuropeptide Y (NPY) and increased expression of the anorexigenic proopiomelanocortin (POMC) in the hypothalamus, being a plausible physiological pathway, as mRNA of expressed vaspin was found in cerebrospinal fluid [157]. Another effect of central administration of vaspin in rats is the inhibition of hepatic glucose synthesis and improved glucose infusion, most likely via the hepatic branch of the vagus nerve, thereby promoting euglycemia in the high-fat diet [159]. As insulin has a satiating effect, a different basis of vaspin’s anorectic mechanism may be due to its influence on prolonging insulin half-time via inhibition of KLK7, responsible for insulin degradation [18,114]. Interestingly, vaspin serum levels are highest right before a meal, when insulin serum levels are rather low [97], which may prove its physiological importance. Vaspin, a serpin family member and apoptosis inhibitor, exhibits a rapid decline in serum concentration following its action, likely due to its role in protecting insulin molecules [1,18,116]. 

In contrast to rats, mice vaspin expression is more pronounced in the liver than in the WAT, where its synthesis is induced by plasma insulin level elevation and strictly depends on it [160]. This may be another important clue about vaspin’s exact role, as it corresponds with earlier findings about its relationship with insulin. Unfortunately, it cannot be conclusively stated that vaspin reduces hunger, as a human study reported a somewhat opposite effect: higher serum vaspin levels were associated with increased feelings of hunger [161]. This observation is consistent with findings that vaspin levels rise prior to expected meals [97]. In contrast to these studies, a three-day fast was found to not to affect vaspin concentration in humans [127]. With these interspecific contradictions, further studies examining the effect of vaspin administration in people are needed to establish certain conclusions on vaspin effects on human eating habits.

Vaspin likely plays a role in appetite regulation, yet dietary intake also impacts serum vaspin concentrations. While healthy diet combined with training results in higher vaspin levels despite weight loss [63], diet alone can also affect training results, as vaspin levels in rats were raised after training-induced mass reduction when their diet consisted mainly of sugars. Such correlation was not observed in rats fed with high-fat chow [64]. While rodents exhibit increased vaspin levels on a high-fat diet, human children on a ketogenic diet for epilepsy show significantly lower vaspin concentrations [162]. Next, an example of dietary impact on vaspin is the Mediterranean diet combined with time-restricted eating (two times a day, at 8 a.m. and 8 p.m.), which decreased serum levels, while a single time-restricted meal did not cause such an effect [67,163].

### 4.3. Type 1 Diabetes Mellitus

Type 1 diabetes mellitus (T1DM), similarly to T2DM, causes impairment of blood glucose level regulation and therefore hyperglycemia [164]. However, the basis of this state is much different from T2DM. Pancreatic β-cells are destroyed by the immunological compartment, leading to complete insulin deficiency, which, without medical treatment, inevitably results in diabetic ketoacidosis and death [164]. Studies exploring the function of vaspin in T1DM may provide information on the potential association between it and immunology, as well as additional proof of its compensatory role in counteracting pancreatic β-cell degradation. Unfortunately, this field is still relatively new, and the limited number of available studies—many of which report contradictory findings—precludes drawing any definitive conclusions. In 2015 it was reported that vaspin was increased in adults with T1DM [165]. Next, a 2020 experiment on T1DM-induced mice showed lowered vaspin concentrations [166], and recently, a study performed on children with T1DM in 2024 demonstrated no correlation at all [167]. More studies are required, as immunologic cells could be another target of vaspin action and might open new perspectives regarding vaspin’s role.

## 5. Discussion

Over the past two decades, adipokines have emerged as highly attractive research targets, and the volume of studies investigating them remains substantial. Since its discovery 20 years ago, vaspin has received considerable attention and is now recognized as an important factor in human metabolism and a significant regulator of homeostasis [17,30,114]. However, many aspects of vaspin’s function remain elusive. The primary aim of our review was not only to summarize current knowledge on vaspin but also to propose new perspectives and highlight the challenges associated with research on this adipokine.

The main research focus was on T2DM and obesity, as vaspin, as an adipokine, was suspected to play a significant role in their pathogenesis [1,54,168]. Multiple experiments in rats and mice lacking the ability to synthesize vaspin demonstrated that the absence of this adipokine predisposes animals to the development of T2DM and obesity [33], whereas its administration ameliorates these conditions [1]. Unfortunately, a study on a human population with the rare mutation rs61757459, causing a reduction in serum vaspin levels, could not confirm the same observation due to the very small study sample [169]. Nonetheless, a large number of studies describe vaspin as a compensative hormone in MetS [1,54], particularly given that its initial characterization was performed in OLETF rats—an established animal model of human MetS [55,170].

According to published studies, vaspin can be associated with multiple physiological and pathophysiological conditions. Its concentrations are increased in obesity [2,4,56,57,58], T2DM [56,95,96], PCOS [120,121,122] and MetS [1,54]. Higher vaspin levels are positively correlated with triglycerides and, to a lesser extent, LDL-cholesterol [63,75,76], improved outcomes following bariatric surgery [88] and the presence of coronary artery disease [148,150]. At the same time, vaspin appears to exert protective effects against hepatic fibrosis [38], dyslipidemia [33,74], weight gain [71], hyperinsulinemia [71], atherogenesis [35,77,110,135,146,147] and inflammation [1,30,92]. Conversely, vaspin concentrations are associated with myocardial infarction [155,156] and severe course of ischemic stroke [152,153,154]. Despite existing knowledge regarding these relationships, the development of standardized reference ranges or diagnostic cut-off values remains highly restricted. This limitation arises mainly from the relatively small sample sizes of available studies, with only a few meta-analyses conducted to date [56,171,172].

Initially, Hida et al. characterized vaspin as an “insulin-sensitizing” adipokine, as it reduced blood glucose levels without altering insulin concentration [1]. Since then, numerous molecular pathways and sites of action for vaspin have been described. Vaspin inhibits KLK7 and KLK14 [18,19], therefore stabilizing insulin half-time [18,21,28] and reducing infiltration of inflammatory macrophages [29], resulting in decreased plaque formation [146,147]. Vaspin inhibits NF-κB, consequently reducing expression of TNF-α, IL-6, IL-1 and inflammation pathways [92,137]. These actions, among others, help prevent endothelial damage in MetS [110,135] and enhance insulin sensitivity and the functional integrity of pancreatic β-cells in T2DM [30]. In the pancreas, vaspin upregulates IRS-2 expression, thereby promoting β-cell proliferation and alleviating insulin resistance [30]. Additionally, it increases NO bioavailability [136,139] and influences various tissues via GRP78 [33,114]. In endothelial cells, this includes reducing apoptosis and suppressing NF-κB activity. Vaspin also acts on the nervous system, influencing eating behavior and parasympathetic innervation [157,158,159]. Despite its pleiotropic effects, vaspin primarily modulates insulin resistance [72,73] in both T2DM and obesity, counteracting the various components of MetS.

Given these effects, vaspin has been proposed as a potential therapeutic agent for metabolic diseases, as well as a biomarker for monitoring treatment efficacy [173,174]. However, it is important to point out that few studies have investigated vaspin administration in human patients, and its exact effects, timing of onset and safety profile remain unclear. This issue is particularly relevant given the often divergent results observed between rodent models and humans. Moreover, there is a substantial lack of longitudinal studies assessing vaspin concentrations in patients undergoing treatment; notably, no such studies have been conducted in individuals with T2DM [69,73,85,86,117,120]. Long-term observation of untreated patients is practically unfeasible due to ethical considerations.

Despite its overwhelmingly positive reputation as a compensatory and protective protein, some studies suggest that vaspin may have potentially harmful effects [87,175], which makes clinical research involving humans particularly challenging. Nevertheless, without such studies, further progress in understanding vaspin’s role and therapeutic potential will remain limited.

An additional and significant challenge in vaspin research is the large number of contradictory findings across its various domains. For nearly every study, there appears to be another reporting opposing results. To illustrate this issue, we compiled a table summarizing most of the relevant works in Appendix A, particularly where controversies exist. A major obstacle is the lack of standardization, which hampers consistent progress. Each original study employs its own sampling protocols and measurement methodologies, and even meta-analyses highlight substantial limitations due to extensive exclusions. Consequently, it remains difficult to determine whether these contradictions arise from ethnic differences, methodological inconsistencies or simply small sample sizes.

## Figures and Tables

**Figure 1 biomedicines-13-03040-f001:**
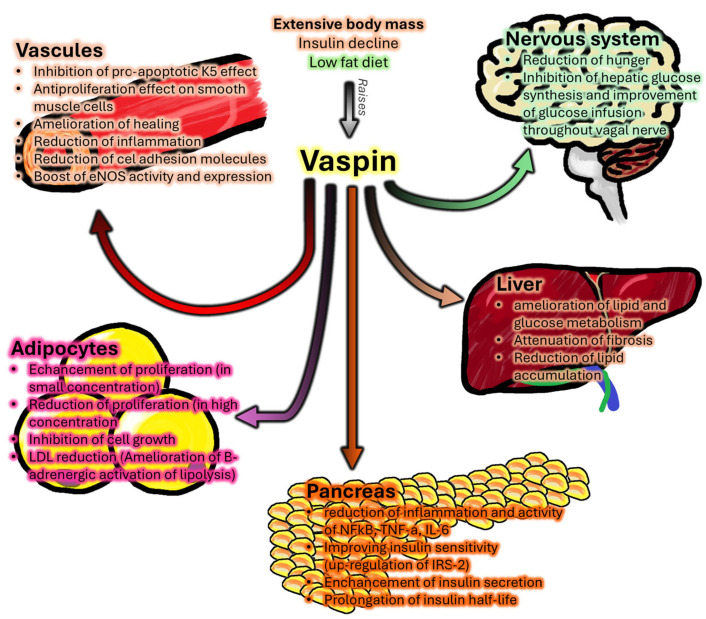
A summary of the tissue-specific biological activities of vaspin in the human body.

**Figure 2 biomedicines-13-03040-f002:**
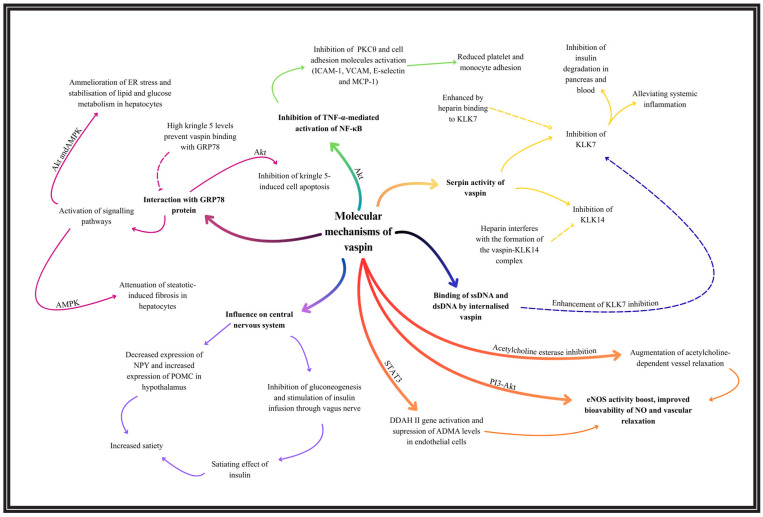
A brief overview of selected molecular mechanisms of vaspin.

**Table 1 biomedicines-13-03040-t001:** Metabolic syndrome diagnosis criteria.

Diagnostic Parameter	Men	Women
Waist circumference	>40 inches	>35 inches
Serum triglycerides level	>150 mg/dL	
High-density lipoprotein cholesterol	<40 mg/dL	<50 mg/dL
Fasting glucose	>100 mg/dL	
Blood pressure values	>130/85 *	

*—either systolic, diastolic or both.

## Data Availability

Documents containing all extracted data are available in the manuscript.

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
