# Peer review of "Biomedicines2025, 13(12), 3040;https://doi.org/10.3390/biomedicines13123040"

_biomedicines, 2025, doi:10.3390/biomedicines13123040_

Round 1

Reviewer 1 Report

Comments and Suggestions for Authors

The review is overall well written, quite comprehensive, and supported by a large number of up-to-date references. The topic is presented clearly and the authors have successfully integrated findings from both basic and clinical research.

I recommend adding a more explicit section highlighting the current gaps in the literature, particularly regarding the SERPINA12–KLK7 axis and its role in adipose–metabolic–cardiovascular crosstalk. In my opinion, two major points deserve stronger emphasis:

    • the lack of large longitudinal studies measuring circulating vaspin over time and linking it to clinically relevant endpoints;

    • the absence of validated diagnostic or prognostic cut-offs, which currently limits the translational relevance of vaspin as a biomarker.

Comments on the Quality of English Language
  • Please check the text for minor inaccuracies. For example, in Appendix A the protein is referred to as “waspina”, which appears to be a typographical error.

  • The sentence on line 522 in the Discussion is difficult to follow and should be rephrased for clarity.

Author Response

Comment 1: I recommend adding a more explicit section highlighting the current gaps in the literature, particularly regarding the SERPINA12–KLK7 axis and its role in adipose–metabolic–cardiovascular crosstalk. In my opinion, two major points deserve stronger emphasis:

    • the lack of large longitudinal studies measuring circulating vaspin over time and linking it to clinically relevant endpoints;
    • the absence of validated diagnostic or prognostic cut-offs, which currently limits the translational relevance of vaspin as a biomarker.

Response 1: Thank you for highlighting these research gaps. Regarding the SERPINA7–KLK7 axis, we aimed to present the entirety of current knowledge, which makes it difficult to identify specific research gaps. Nevertheless, we highlighted the need for further investigation due to the mutually exclusive conclusions reported in the literature, as noted in lines 109–111.

In response to your further suggestions, these important issues have been discussed in added lines 610-614 and 586-589.  If any further adjustments are needed in this regard, please let us know.

Comment 2: Please check the text for minor inaccuracies. For example, in Appendix A the protein is referred to as “waspina”, which appears to be a typographical error.

Response 2: Thank you for pointing this out, it was indeed an typographical error and has been appropriately corrected.

Comment 3: The sentence on line 522 in the Discussion is difficult to follow and should be rephrased for clarity.

Response 3: As suggested, lines 605-606 (previously 522-524) were rephrased and now read as follows: “Given these effects, it is strongly suggested that vaspin may, in the future, be used as a medication in metabolic diseases and as a marker for monitoring treatment efficacy.”

Reviewer 2 Report

Comments and Suggestions for Authors

The review is timely, well structured and represents a comprehensive overview of the topic. Literature presented is relevant to a broad spectrum of specialties including but not limited to endocrine, metabolic and cardiovascular disease. However, the text is overly long and redundant in some sections and would definitely benefit from the inclusion of conceptual scheme that link multiple observations together (as opposed to a boring list of research studies) to make it shorter and more readable. 

These are some comments to enhance flow, presentation and overall readability of the manuscript: 

  • The authors use “vaspin” and “SerpinA12” sometimes within the same section, the inconsistent switching could confuse readers even if they refer to the same molecule, It is better to choose one primary term (vaspin) and use it consistently throughout the manuscript and introduce the synonym once in the introduction and then use the preferred name uniformly throughout the manuscript.
  • In the abstract: “However, despite intensive research on serpin over the past 20 years”, does this refer to the serpin family in general, or to vaspin?
  • The Introduction provide important background info about vaspin but it lacks coherent flow and seems to jump between discovery, metabolic relevance, detailed protein structure, and tissue expression without a guiding flow. It also lacks an explanation of the research gaps or the rationale for the review. I suggest reorganizing the introduction to provide a more logical flow and moving structural details to the second section.
  • I suggest changing the title of Section 3 “Vaspin in obesity-related diseases” which does not seem to fully match the content. Although this section discusses obesity-related diseases, it also includes satiety (This is a physiological mechanism, not a disease), and type 1 diabetes (This is autoimmune, not obesity related in any way). So the title should reflect the broad physiological and pathophysiological roles of vaspin that are discussed in this part of the manuscript.
  • Meanwhile, Cardiovascular disease are placed in a separate section, although it also can be connected to obesity, just like other disease placed under section 3, it might be more clearer to make each subsection in Section 3 its own main section or adjust the section title so it accurately reflects the broader scope, which help provide rational for making cardiovascular section separate.
  • It is also a good idea to make the table, follow the flow of the manuscript (sections and subsections. :Obesity, Diabetes, CVD… or even split into multiple tables.
  • A schematic that summarizer vaspin molecular mechanism and signaling pathways would greatly enhance the clarity and provide visual summary of the section. Another figure that show vaspin’s activity across tissues would be helpful for the readers.

Author Response

Comment 1: However, the text is overly long and redundant in some sections and would definitely benefit from the inclusion of conceptual scheme that link multiple observations together (as opposed to a boring list of research studies) to make it shorter and more readable. 

Response 1: We agree that a scheme grouping results into categories connected to various organs would be attractive for readers. Thus, we compiled Figure 1 performing this function located on page 2. This addition will help readers quickly familiarise themselves with the issues discussed in the article.

Comment 2: The authors use “vaspin” and “SerpinA12” sometimes within the same section, the inconsistent switching could confuse readers even if they refer to the same molecule, It is better to choose one primary term (vaspin) and use it consistently throughout the manuscript and introduce the synonym once in the introduction and then use the preferred name uniformly throughout the manuscript.

Response 2: Thank you for pointing this out, we read through the article to address this issue. Such corrections were made in lines 23 and 46.

Comment 3: In the abstract: “However, despite intensive research on serpin over the past 20 years”, does this refer to the serpin family in general, or to vaspin?

Response 3: This phrase was intended to refer to the early history of vaspin research, beginning with its discovery in 2005. Since the wording was imprecise, we replaced “serpin” with “vaspin” in line 28.

Comment 4: The Introduction provide important background info about vaspin but it lacks coherent flow and seems to jump between discovery, metabolic relevance, detailed protein structure, and tissue expression without a guiding flow. It also lacks an explanation of the research gaps or the rationale for the review. I suggest reorganizing the introduction to provide a more logical flow and moving structural details to the second section.

Response 4: Thank you for suggesting this - we agree that clearing the introduction section will be beneficial for the article. To achieve this, we moved the portion describing the molecular details of vaspin (lines 50–55) from the Introduction to subsection 2.1. The serpin activity of vaspin, keeping the text unchanged. Additionally, the Introduction has been expanded with several lines briefly outlining the major research challenges and the rationale for this review.

Comment 5: I suggest changing the title of Section 3 “Vaspin in obesity-related diseases” which does not seem to fully match the content. Although this section discusses obesity-related diseases, it also includes satiety (This is a physiological mechanism, not a disease), and type 1 diabetes (This is autoimmune, not obesity related in any way). So the title should reflect the broad physiological and pathophysiological roles of vaspin that are discussed in this part of the manuscript. Meanwhile, Cardiovascular disease are placed in a separate section, although it also can be connected to obesity, just like other disease placed under section 3, it might be more clearer to make each subsection in Section 3 its own main section or adjust the section title so it accurately reflects the broader scope, which help provide rational for making cardiovascular section separate.

Response 5: Thank you very much for this comprehensive comment – fully agreed. To address this issue, we decided to reorganize the order of sections and subsections. In Section 3: “Vaspin in obesity-related diseases”, we retained only those subsections that directly concern metabolic disorders: obesity, metabolic syndrome, dyslipidaemia, type 2 diabetes mellitus, and polycystic ovary syndrome — the latter being strongly associated with metabolic dysfunction. The remaining subsections were moved to a newly created Section 4: “Vaspin in a wider context”, as they either represent sequelae of obesity (e.g., cardiovascular diseases) or are not related to obesity at all (e.g., type 1 diabetes mellitus). If any further adjustments are needed in this regard, please let us know.

Comment 6: It is also a good idea to make the table, follow the flow of the manuscript (sections and subsections. :Obesity, Diabetes, CVD… or even split into multiple tables.

Response 6: As suggested, we arranged Appendix A rows in order of subsection appearance in the text.

Comment 7: A schematic that summarizer vaspin molecular mechanism and signaling pathways would greatly enhance the clarity and provide visual summary of the section. Another figure that show vaspin’s activity across tissues would be helpful for the readers.

Response 7: We agree such schematic would greatly improve the article, providing adequate summary of second section and selected mechanisms described in the later sections. Thus, we compiled Figure 2 on page 5. Figure 1 mentioned in Response 1 explains effect of vaspin on various tissues.

Round 2

Reviewer 1 Report

Comments and Suggestions for Authors

Thank you for your revision of this manuscript. I appreciate the time that you have put into revising your manuscript based on my comments.

Comments on the Quality of English Language
  • Please check the text for minor inaccuracies. For example, in Appendix A the protein is referred to as “waspina”, which appears to be a typographical error.

  • The sentence on line 522 in the Discussion is difficult to follow and should be rephrased for clarity.

Author Response

Comment 1: Please check the text for minor inaccuracies. For example, in Appendix A the protein is referred to as “waspina”, which appears to be a typographical error.

The sentence on line 522 in the Discussion is difficult to follow and should be rephrased for clarity.

Response 1: Both issues were addressed in the previous round of revision and have been corrected accordingly. “Waspina” was indeed a typographical error and has now been corrected. Lines 611–612 (previously 522–524) have been rephrased and now read as follows: “Given these effects, vaspin has been proposed as a potential therapeutic agent for metabolic diseases, as well as a biomarker for monitoring treatment efficacy.”

Finally, we have carefully reviewed the entire manuscript and implemented necessary orthographic and stylistic corrections to improve the overall quality and readability of the text.

Reviewer 2 Report

Comments and Suggestions for Authors

The authors have adequately responded to most of the comments raised in the first revision cycle, however, the figures added to the revised version (although conceptually sound) are lacking in terms of legibility and visual clarity especially figure 2. 

These are my comments:

  1. For figure 1 (maintain the same scheme and information) but reproduce the figure so that the font overlaying the organ images is legible (currently it is not and is of sub-standard resolution.
  2. Figure 2 is overly crowded, suffers from poor resolution and does not satisfy the reason behind its inclusion. 

Author Response

Comment 1: For figure 1 (maintain the same scheme and information) but reproduce the figure so that the font overlaying the organ images is legible (currently it is not and is of sub-standard resolution.

Response 1: Thank you for your suggestions. We have adjusted the text in Figure 1 to improve its legibility. We also ensured that the image resolution meets the required standards, as explained in greater detail in Response 2.

Comment 2: Figure 2 is overly crowded, suffers from poor resolution and does not satisfy the reason behind its inclusion. 

Response 2: Similarly to Figure 1, we reassessed the resolution of Figure 2. The resolution of both figures is very high: the attached PNG files have dimensions of 2248 × 2000 pixels and 3223 × 2160 pixels, respectively - values that approach 4K resolution (3840 × 2160). For comparison, the average monitor displays 1920 × 1080 pixels. Combined with a DPI of 600, this confirms that the graphics fully satisfy the journal’s requirements for high-definition images stated in guidelines for authors.

It is likely that the reduced quality you are observing results from Microsoft Word’s automatic image preview compression. Please note that the initially seemingly blurry schematic becomes sharper upon zooming in. This indicates that the resolution of both figures is indeed sufficient and should not pose any issue in the online publication.

Additionally, in Figure 2—whose inclusion was suggested during the first round of review—we reduced content density by decreasing the number of words while preserving the intended meaning. This figure summarizes complex molecular mechanisms of vaspin and its associated signalling pathways. All known mechanisms have been included, and the figure reflects the maximum degree of detail that can be reliably and clearly presented within a single illustration. If you have any specific improvements in mind, we kindly ask you to elaborate on them.